# Time intervals and distances travelled for prehospital ambulance stroke care: data from the randomised-controlled ambulance-based Rapid Intervention with Glyceryl trinitrate in Hypertensive stroke Trial-2 (RIGHT-2)

Mark Dixon [1,2] Jason P Appleton,[3] Polly Scutt,[1] Lisa J Woodhouse [1]
Lee J Haywood,[1] Diane Havard,[1] Julia Williams [4]
Aloysius Niroshan Siriwardena [5] Philip M Bath,[1] on behalf of the RIGHT-2 Investigators

**Correspondence to**
Philip M Bath;
philip.bath@nottingham.ac.uk

## ABSTRACT

**Objectives** Ambulances offer the first opportunity to evaluate hyperacute stroke treatments. In this study, we investigated the conduct of a hyperacute stroke study in the ambulance-based setting with a particular focus on timings and logistics of trial delivery.

**Design** Multicentre prospective, single-blind, parallel group randomised controlled trial.

**Setting** Eight National Health Service ambulance services in England and Wales; 54 acute stroke centres.

**Participants** Paramedics enrolled 1149 patients assessed as likely to have a stroke, with Face, Arm, Speech and Time score (2 or 3), within 4 hours of symptom onset and systolic blood pressure >120 mm Hg.

**Interventions** Paramedics administered randomly assigned active transdermal glyceryl trinitrate or sham.

**Primary and secondary outcomes** Modified Rankin scale at day 90. This paper focuses on response time intervals, distances travelled and baseline characteristics of patients, compared between ambulance services.

**Results** Paramedics enrolled 1149 patients between September 2015 and May 2018. Final diagnosis: intracerebral haemorrhage 13%, ischaemic stroke 52%, transient ischaemic attack 9% and mimic 26%. Timings (min) were (median (25–75 centile)): onset to emergency call 19 (5–64); onset to randomisation 71 (45–116); total time at scene 33 (26–46); depart scene to hospital 15 (10–23); randomisation to hospital 24 (16–34) and onset to hospital 97 (71–141). Ambulances travelled (km) 10 (4–19) from scene to hospital. Timings and distances differed between ambulance service, for example, onset to randomisation (fastest 53 min, slowest 77 min; p<0.001), distance from scene to hospital (least 4 km, most 20 km; p<0.001).

**Conclusion** We completed a large prehospital stroke trial involving a simple-to-administer intervention across multiple ambulance services. The time from onset to randomisation and modest distances travelled support the applicability of future large-scale paramedic-delivered

## STRENGTHS AND LIMITATIONS OF THIS STUDY

⇒ The first multicentre paramedic-delivered ambulance-based randomised controlled trial in stroke in the UK.

⇒ Ambulance response time intervals and distances are collated and reported for 1149 patients assessed as likely to have a stroke.

⇒ The time interval between arrival at hospital and the ambulance becoming available for the next emergency call (hospital turnaround) is not captured, but worth considering for future trials.

⇒ Timing and logistic data may not be fully representative of all urban and rural locations due to non-participation of some hospitals and ambulance stations within ambulance service regional areas.

ambulance-based stroke trials in urban and rural locations.

**Trial registration number** ISRCTN26986053.

## INTRODUCTION

Routine prehospital management of suspected acute stroke involves rapid identification of suspected stroke using a validated stroke screening tool, prompt transport, pre-arrival notification and primary stabilisation to the nearest appropriate receiving stroke centre.[1] The mainstays for hyperacute management of stroke in hospital include urgent neuroimaging, stroke unit care, reperfusion therapy for ischaemic stroke and blood pressure (BP) lowering for intracerebral haemorrhage.[2] For reperfusion therapies, shortening the time from symptom onset to treatment improves functional outcome and this has become the

aim of prehospital and in-hospital acute stroke services.[3–6] Thus, ambulance services play a crucial role in assessing, identifying and conveying patients with suspected stroke to primary and comprehensive stroke centres, which may include bypassing local emergency departments.

Timely prehospital care for stroke is dependent on several factors that include rapid recognition of potential stroke and calling for help,[7] ambulance response times encompassing symptom onset to arrival at hospital,[8] distance from scene to hospital[9] and the accuracy of identifying patients with true stroke or transient ischaemic attack from those with a stroke mimic.[10] There are a small, but growing number of studies that explore randomised paramedic-initiated interventions commencing in the ambulance for acute stroke. However, few studies have systematically analysed these parameters and the factors that influence them in acute prehospital stroke practice.[4 9 11 12]

Here, we report the logistics underlying patient recruitment to the Rapid Intervention with Glyceryl trinitrate in Hypertensive stroke Trial-2 (RIGHT-2), a large ambulance-based stroke trial in the UK that investigated the efficacy of transdermal glyceryl trinitrate as a paramedic-delivered intervention in suspected acute stroke. Specifically, ambulance response times and distance travelled across multiple organisations in this setting are assessed.

## METHODS

### RIGHT-2 trial

RIGHT-2 commenced recruitment in September 2015 with the first participant recruited on 22 October 2015.

RIGHT-2 was a multicentre prospective, single-blind, parallel group randomised controlled trial; the protocol, statistical analysis plan, baseline data, main results and subgroup results in participants with a final diagnosis of intracerebral haemorrhage are published.[13–17] Briefly, adult patients with suspected stroke presenting to the emergency service via an emergency call were recruited if they: were FAST-positive (facial weakness, arm weakness, speech abnormality; with test score 2 or 3), had systolic BP of >120 mm Hg, were within 4 hours of symptom onset, presented to trial-trained paramedics from eight UK ambulance services and were to be taken to a trial-participating hospital. Patients were randomised to receive transdermal glyceryl trinitrate (GTN) or sham patch in the ambulance and this was continued for three further days during hospital admission.[13] The study was undertaken across eight UK ambulance services (AS): East of England AS (EEAS), East Midlands AS (EMAS), London AS (LAS), South-Central AS (SCAS), South-West AS (SWAS), Welsh AS (WAS), West-Midlands AS (WMAS) and Yorkshire AS (YAS). All participating ambulance services used FAST identification and protocols consistent with national guidelines.

For each eligible patient, the enrolling paramedic assessed capacity and obtained patient or proxy consent (from a relative on the scene, or from the paramedic witnessed by a colleague), completed a written case report form to capture in-ambulance baseline and on-treatment data and applied the transdermal patch of GTN or sham dressing.[13] Ambulance-related data not recorded at source were confirmed by research paramedics from participating ambulance services after review of control room timing logs or patient care records, and then entered into the trial database.

### Timings and distances

Timings were obtained from each ambulance service (time of emergency call, resource dispatch, scene arrival and departure, hospital arrival) and from paramedic records (consent for trial enrolment, randomisation, application of study treatment). Paramedic-documented history provided the time of symptom onset or, where unclear, the last known well time.

Distance measurements were calculated from the address or postcode of the emergency location, where available, to the expected stopping point for the ambulance at the destination hospital (accident and emergency or stroke unit entrance) to the nearest 10 metres using Google Maps; one ambulance service was unable to provide postcode information due to time constraints. One ambulance service was able to provide the linear distance from the location of the ambulance at the point of dispatch to the scene of the emergency.

A comparison of urban versus rural ambulance services arbitrarily divided ASs by <25% rural versus >25% rural (as defined in table 1; online supplemental table I).

### Comparison of trial and non-trial patients

One ambulance service provided response time interval and distance data for a cohort (n=49) of patients with confirmed stroke who were not enrolled into RIGHT-2 (attended by non-trial trained paramedics) but were transported to the same specialist stroke centres participating in the trial.

### Statistical analysis

Time intervals (in min), distances (in km) and baseline characteristics were compared between ambulance services using $\chi^2$ and Kruskal-Wallis (one-way analysis of variance on ranks) tests. Multiple comparison procedures (Dunn's with Bonferroni correction) were used to assess which ambulance service differed from the others. Spearman and point-biserial correlations were performed to identify the relationship between baseline variables, times and distances. Data are number (%), median (IQR) or mean (SD). Statistical significance was defined overall at p<0.05, and at p<0.001 for correlation matrices and multiple comparisons. Statistical analyses were conducted with SPSS V.24 (IBM, New York, USA).

### Patient and public involvement

This study was supported by public members of the trial steering committee who were involved throughout, including in trial design, development, conduct, periodic review and dissemination of results.

**Table 1** Characteristics of participating ambulance service as of 31 May 2018. Data are numbers (%)

| | E&W | EEAS | EMAS | LAS | SCAS | SWAS | WAS | WMAS | YAS |
|---|---|---|---|---|---|---|---|---|---|
| Time in trial (months) | 32 | 27 | 32 | 14 | 4 | 27 | 22 | 14 | 29 |
| Patients | 1149 | 178 | 218 | 202 | 7 | 265 | 89 | 37 | 153 |
| Participating hospitals | 54 | 5 | 10 | 3 | 1 | 13 | 4 | 5 | 13 |
| Area (km$^2$) | 122065 | 19424 | 16710 | 1605 | 9204 | 25899 | 20735 | 12949 | 15539 |
| Population | | | | | | | | | |
| Overall (×1000) | 53000 | 5800 | 4800 | 8600 | 7000 | 5300 | 2900 | 5600 | 5338 |
| Living in rural areas* (%) | 17.6 | 28.9 | 26.7 | 0.2 | 20.4 | 31.6 | 32.8 | 15.1 | 17.5 |
| Strokes (/year)† | 90781 | 9145 | 9246 | 13118 | 7763 | 10442 | 7400 | 8701 | 7931 |
| Adjusted ratio /1000 | 1.71 | 1.58 | 1.92 | 1.52 | 1.11 | 1.97 | 2.55 | 1.55 | 1.49 |
| Call volume (/day) | 24661 | 2800 | 2500 | 5193 | 1479 | 3077 | 1331 | 3000 | 2336 |
| Participating ambulance stations | 270 | 24 | 50 | 23 | 3 | 73 | 34 | 17 | 63 |
| Paramedics employed | 22000 | 2000 | 1111 | 2864 | 1780 | 1788 | 1310 | 1300 | 1592 |
| Trained in RIGHT-2 | 1492 | 145 | 193 | 325 | 63 | 313 | 165 | 124 | 142 |
| Paramedics who recruited | 516 | 58 | 75 | 120 | 6 | 112 | 47 | 23 | 75 |
| Patients/paramedic | 2.22 | 3.06 | 2.90 | 1.68 | 1.16 | 2.37 | 1.89 | 1.60 | 2.04 |

*2011 Census.[18]
†Number of patients with suspected stroke assessed face-to-face 2015/2016.
EEAS, East of England Ambulance Service NHS Trust; EMAS, East Midlands Ambulance Service NHS Trust; E&W, England and Wales; LAS, London Ambulance Service; RIGHT-2, Rapid Intervention with Glyceryl trinitrate in Hypertensive stroke Trial-2; SCAS, South-Central Ambulance Service NHS Foundation Trust; SWAS, South-West Ambulance Service NHS Foundation Trust; WAS, Welsh Ambulance Service NHS Trust; WMAS, West-Midlands Ambulance Services; YAS, Yorkshire Ambulance Service NHS Trust.

## RESULTS

RIGHT-2 recruited 1149 patients between September 2015 and May 2018. Table 1 outlines patient recruitment across the various participating ambulance services, which collectively covered an area of 122065 km$^2$ in England and Wales (ie, 42% of the land area of these countries). Ambulance services varied considerably in size (1605 km$^2$ vs 25899 km$^2$), population served per service (2.9 million vs 8.6 million)[18] and annual stroke events (7400 vs 13 118) (table 1). Altogether 1492 paramedics volunteered to be trained in the trial, of whom 516 (36%) recruited at least one patient. Where two or more trial paramedics were present at the scene, the paramedic initiating randomisation was credited. On average, 2.2 patients were recruited by each paramedic who enrolled at least one patient although this varied between ambulance services (1.1 vs 3.1).

### Patient characteristics

Of the 1149 patients recruited, average age was 73 (15) years, women 48%, BP 162 (25)/92 (18) mm Hg, Glasgow Coma Scale (GCS) 13.9 (1.7) and FAST score of three 60% (online supplemental table II). The final diagnosis varied between ambulance services, with the rate of conditions mimicking acute neurovascular disease ranging from 14.3% to 36.1%. This is consistent with other prehospital trials without physician presence or mobile stroke unit care, and the rate of stroke mimic reported here is explored elsewhere.[19] Baseline temperature also varied. Otherwise, baseline characteristics did not differ

between ambulance service. As age increased, BP and glucose were higher, and heart rate, FAST and GCS lower (online supplemental table III). Informed consent was provided by 603 (53%) patients, 431 (38%) relatives and 115 (10%) paramedics witnessed by a colleague on scene.

### Time intervals

The time intervals for various stages in the journey from stroke scene to hospital are shown in online supplemental table IV. Overall, the median time from symptom onset to emergency call was 19 (IQR 5–64) min and this did not differ between ambulance services (online supplemental tables IV and V). The median time from emergency call to ambulance dispatch was 3 (1–7) min and varied between ambulance service (1 min vs 5 min). An ambulance resource arrived at the scene within 8 (5–13) min from being dispatched (and 10 (6–16) minutes if only including RIGHT-2 trained paramedics) with this varying between ambulance service (8 min vs 12 min).

The median time from onset of symptoms to randomisation was 71 (45–116) minutes (table 2, figure 1) and this varied between ambulance service (53 min vs 77 min). Significantly, randomisation occurred within 30 and 60 min of symptom onset in 104 (9.1%) and 491 (42.9%) participants, respectively (table 2). Ambulance resources spent a median of 33 (26–46) minutes on scene, though this varied between ambulance services (29 min vs 43 min) (online supplemental table IV). Importantly, time on scene did not differ significantly when comparing RIGHT-2 patients vs non-RIGHT-2 patients 34 (26–44) and

**Table 2** Timings: symptom onset to randomisation (OTR) (min). Data are N (%), median (25–75 centile); comparison by Kruskal-Wallis test

| Min | E&W | EEAS | EMAS | LAS | SCAS | SWAS | WAS | WMAS | YAS | p |
|---|---|---|---|---|---|---|---|---|---|---|
| OTR | | | | | | | | | | |
| N (%) | 1149 | 178 (15.5) | 218 (19.0) | 202 (17.6) | 7 (0.6) | 265 (23.1) | 89 (7.7) | 37 (3.2) | 153 (13.3) | |
| Median (25–75 centile) | 71 (45–116) | 73 (47–120) | 59 (35–100) | 77 (51–124) | 53 (45–65) | 75 (49–107) | 75 (48–123) | 60 (32–115) | 70 (45–118) | 0.001 |
| N (%) | | | | | | | | | | <0.001 |
| ≤30 | 104 (9.1) | 15 (8.4) | 38 (17.4) | 11 (5.4) | 1 (14.3) | 16 (6.0) | 7 (7.9) | 6 (16.2) | 10 (6.5) | |
| 31–60 | 387 (33.8) | 63 (35.4) | 82 (37.6) | 61 (30.2) | 3 (42.9) | 82 (30.9) | 28 (31.5) | 13 (35.1) | 56 (36.6) | |
| 61–90 | 258 (22.5) | 32 (18.0) | 34 (15.6) | 51 (25.2) | 0 (0.0) | 77 (29.1) | 19 (21.3) | 5 (13.5) | 38 (24.8) | |
| 91–120 | 136 (15.1) | 25 (14.0) | 19 (8.7) | 25 (12.4) | 0 (0.0) | 36 (13.6) | 13 (14.6) | 5 (13.5) | 13 (8.5) | |
| 121–180 | 173 (15.1) | 30 (16.9) | 33 (15.1) | 28 (13.9) | 0 (0.0) | 40 (15.1) | 16 (18.0) | 6 (16.2) | 20 (13.1) | |
| 181–240 | 76 (6.6) | 12 (6.7) | 9 (4.1) | 20 (9.9) | 0 (0.0) | 13 (4.9) | 5 (5.6) | 2 (5.4) | 15 (9.8) | |
| >240 | 15 (1.2) | 1 (0.5) | 3 (1.4) | 6 (3.0) | 1 (14.3) | 1 (0.4) | 1 (1.1) | 0 (0.0) | 2 (0.7) | |

EEAS, East of England Ambulance Service NHS Trust; EMAS, East Midlands Ambulance Service NHS Trust; E&W, England and Wales; LAS, London Ambulance Service; SCAS, South-Central Ambulance Service NHS Foundation Trust; SWAS, South-West Ambulance Service NHS Foundation Trust; WAS, Welsh Ambulance Service NHS Trust; WMAS, West-Midlands Ambulance Services; YAS, Yorkshire Ambulance Service NHS Trust.

32 (23–41) min, respectively (online supplemental table VI). Transfer time from scene to hospital was a median of 15 (10–23) min, but varied between ambulance service (9 min vs 24 min) (online supplemental table IV). The overall time from symptom onset to arrival at hospital was 97 (71–141) minutes and also varied between ambulance services (86 min vs 109 min) (online supplemental table VII). Time at scene was strongly positively correlated with time from scene to hospital (table 3).

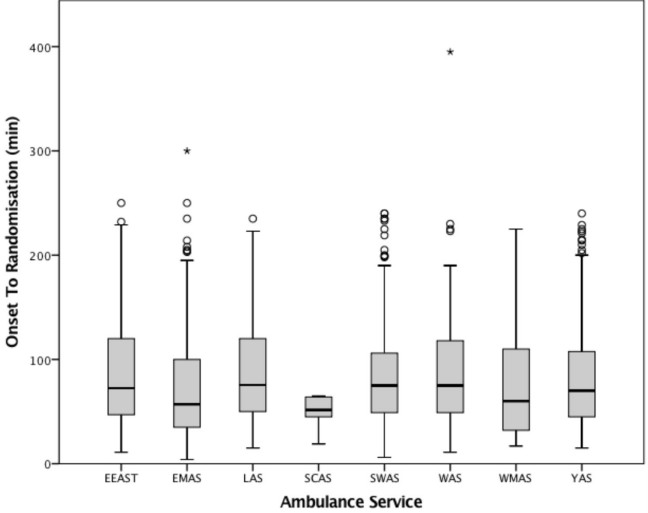

**Figure 1** Box plot of onset to randomisation. EEAS, East of England Ambulance Service NHS Trust; EMAS, East Midlands Ambulance Service NHS Trust; LAS, London Ambulance Service; SCAS, South-Central Ambulance Service NHS Foundation Trust; SWAS, South-West Ambulance Service NHS Foundation Trust; WAS, Welsh Ambulance Service NHS Trust; WMAS, West-Midlands Ambulance Services; YAS, Yorkshire Ambulance Service NHS Trust.

### Distances

The median distance travelled from the postcode of the suspected stroke scene to the receiving hospital was 10.0 (4.4–18.4) km, with considerable variation between ambulance services (4.1 km vs 19.9 km) (online supplemental table VIII). Time from scene to hospital was moderately positively correlated with distance from scene to hospital (online supplemental figure I:A-G present geographical distribution of randomisation by ambulance service).

### Urban versus rural services

When comparing urban and rural ambulance services (online supplemental table I), there was no difference in receipt of the emergency call to dispatching a resource to scene, nor a difference in onset of symptoms to randomisation. The time spent at scene was marginally longer in rural locations and, as anticipated, both conveyance time and distance to the stroke centre was statistically different.

### Comparison of trial and non-trial patients

In the ambulance service with times available for patients not enrolled in the trial, on scene to hospital arrival differed among patients enrolled and not enrolled in RIGHT-2, 10 (0.4–64.7) vs 16 (7.6–24.0) min (online supplemental table VIII). The median distance from dispatch location to scene in the ambulance service with this available (EMAS) was 7.3 km (3.5–12.0).

### DISCUSSION

In this large national prehospital trial, 516 paramedics from eight ambulance services across England and Wales successfully recruited 1149 participants and transported them to 54 hospitals. Paramedics assessed and diagnosed suspected stroke, consented patients and initiated

**Table 3** Univariate Correlation between severity of symptoms, timings and distance from scene to hospital. Data are Spearman's coefficient (p-value)

|  | OTR | FAST | GCS | Scene | OTH | STH | Km |
|---|---|---|---|---|---|---|---|
| OTC | 0.836 (<0.001) | −0.130 (<0.001) | 0.086 (0.003) | 0.05 (0.088) | 0.802 (<0.001) | 0.007 (0.80) | −0.070 (0.033) |
| OTR |  | −0.135 (<0.001) | 0.102 (0.001) | 0.263 (<0.001) | 0.941 (<0.001) | 0.244 (<0.001) | −0.61 (0.61) |
| FAST |  |  | −0.157 (<0.001) | −0.066 (0.026) | −0.133 (<0.001) | −0.046 (0.12) | 0.803 (0.93) |
| GCS |  |  |  | −0.008 (0.80) | 0.115 (<0.001) | 0.084 (0.004) | 0.076 (0.017) |
| Scene |  |  |  |  | 0.326 (<0.001) | 0.791 (<0.001) | 0.104 (0.002) |
| OTH |  |  |  |  |  | 0.403 (<0.001) | 0.216 (<0.001) |
| STH |  |  |  |  |  |  | 0.554 (<0.001) |

FAST, Face, Arm, Speech, Time score; GCS, Glasgow Coma Scale; Km, distance (km) from scene to hospital.; OTC, onset to emergency call; OTH, onset to hospital; OTR, onset to randomisation; Scene, total time spent at scene; STH, time from scene to reach hospital.

randomised treatment. Key timings were: onset to emergency call 19 min, onset to scene 40 min, onset to randomisation 71 min, time at scene 33 min, randomisation to hospital 24 min and depart scene to hospital arrival 15 min; all but the first two differed between ambulance services. The average distance travelled by one ambulance service from dispatch location to scene was 7.3 km and 10.0 km from scene to hospital for all participating ambulance services.

Prehospital time intervals in acute stroke have been described previously,[9] [20–24] but rarely in randomised trials.[4] [11] [12] The symptom onset to randomisation time of 71 min in RIGHT-2 is consistent with two previous UK ambulance-based stroke trials (RIGHT was 55 min and Paramedic Initiated Lisinopril For Acute Stroke Treatment (PIL-FAST) was 70 min)[25] [26] although these were small single centre pilot studies undertaken largely in urban settings. The large US Field Administration of Stroke Therapy - Magnesium trial[27] (FAST-MAG) reported a median of 45 min from symptom onset to receipt of study drug. Nevertheless, these times are all longer than UK multicentre ambulance-based trials outside of stroke, notably the AIRWAYS-2 and PARA-MEDIC-2 trials in cardiac arrest.[28] [29] In PARAMEDIC-2, the onset of symptoms to initiation of treatment in the intervention group was just 21.5 min. The most important driver of this difference is most likely shorter onset to call times for patients who had cardiac arrest than for stroke, and suspected stroke may require more complex assessment both by call handlers and by paramedics on scene. Additional contributors are that cardiac arrest is allocated the highest dispatch priority, an immediate response and patients receive immediate trial treatment with emergency waiver of consent.

The explanation for differences in timings is probably multifactorial but the degree of urban versus rural population is one likely explanation. This was apparent for time spent at the scene and both time and distance to hospital. As expected, there were no differences for receipt of call to dispatch, arrival of RIGHT-2 trained paramedic at scene nor onset of symptoms to randomisation.

There are several strengths of this study. First, RIGHT-2 involved 8 of 11 ambulance services in England and Wales. Of those not participating, two were unable to join because they were involved in another ambulance-based stroke trial[30] and the other involved hospitals that were concerned about adversely impacting on recruitment to commercial trials. Among 1492 trained paramedics in RIGHT-2 procedures 516 consented and randomised a large number of participants, adhered to the protocol and completed specific data recording. It is noted that there are marked differences in recruit numbers between ambulance services. This, in part, is accounted for due to low recruitment during the initial recruitment phase requiring broadening of ambulance services from 5 to 8 and stroke centres from 30 to 54. Furthermore, recruitment hours initially limited to typical working hours for research staff availability were extended to encompass 24/7 recruitment reflective of real-world ambulance care to not limit participation and maximise inclusion. Conflictingly, a small number of stroke centres closed recruitment to ambulances once target numbers of participants had been received and before the end of the recruitment phase highlighting the challenging reliance on dual centres when dealing with research in prehospital stroke.

Second, the consent model applied in RIGHT-2 is unlike any other large-scale ambulance-based studies worldwide to date and builds on previous UK based prehospital stroke pilots.[25] [31] Other prehospital trials in stroke have relied on models of either informed consent,[32] deferred consent[33] or consent by doctor (present or remote).[27] [34] Stroke is complex due to the varying nature of severity of presentations where patients' ability to consent in an informed manner to participate in a clinical study should not be overlooked preserving patient autonomy in accordance with the Declaration of Helsinki.[35] [36] Notwithstanding the complexities of emergency presentations that could impact on decision-making, mental capacity or short intervention windows and the impact these situations bring to truly informed patient consent, the combined consent approach in RIGHT-2 acknowledges

patient autonomy without precluding participation from those who are unable to voice their opinion or who lack presence of a proxy to consent on their behalf.[36] Mechanisms to safeguard consent were built into the protocol through reconfirmation of consent once in hospital for both the prehospital and in-hospital elements, respectively, and patient and public representatives were fully embedded within protocol development and steering group oversight of the trial.[13]

Third, the protocol required flexibility and adaptation to align with individual operational processes specific to each ambulance service to ensure successful delivery of the trial. Fourth, detailed logistic information on timing and distances travelled were collected. Last, the results highlight the successful delivery of a simple, ambulance-based intervention with 43% of the patients receiving the intervention within 2 hours of symptom onset without compromising time on scene required to complete additional research activity.

There are also several study limitations. First, it is recognised that not every receiving stroke unit within each ambulance service region could participate in RIGHT-2 due to capacity and competing research,[30 37] (this included concurrent commercial and post-arrival trials). Therefore, it must be considered that the timing and logistic data of participating hospitals may not be fully representative of all urban and rural locations. However, the intention was not to assess the differences between urban and rural settings, but to shed light on the conduct and deliverability of a prehospital intervention in stroke where time and distance may impede access to specialist stroke services. Furthermore, stroke unit hours of operation varied across the 54 centres with a small number of sites not accepting patients outside working hours which impacted paramedics' decisions to randomise. This reduces the reflection of real-world emergency stroke care. The duration of recruitment varied between regions due to complexities in setting up multicentre research

Additionally, it is acknowledged that the recruitment criteria were broad which resulted in a higher than anticipated proportion of stroke mimics. To mitigate this, mobile stroke unit care is an emerging field where imaging and definitive care delivery at the scene reduces time delays in stroke[38] and could offer improved confidence and precision of diagnosis for prehospital trial enrolment.

Recognising that 516 of 1492 RIGHT-2 trained paramedics (36%) identified and randomised eligible patients, this is consistent with other trials in prehospital stroke.[25] This, in part, is due to the voluntary participation of paramedics in research where records suggest that only one-third of the paramedic workforce participate.[39] Further, in a UK system where response time is one benchmark of the quality of ambulance service provision, ambulance dispatchers are not able to assign specific research-trained personnel to specific emergency calls, instead allocating the nearest available resource to attend. Low recruitment must be considered during the

development of ambulance-based trials and this factor alone has previously resulted in extended recruitment phases, retraining of researchers and extensive study drug availability to achieve preplanned sample sizes.[27 33 40]

Finally, this paper does not capture the time interval between arrival at hospital and handover to the hospital team, nor the time of the ambulance becoming available for the next emergency call (hospital turnaround). During the hospital turnaround period, ambulance staff handover the patient to hospital staff, complete relevant documentation and prepare the vehicle for the next assignment. A rapid hospital turnaround is important for making the vehicle available for waiting emergency calls. While the addition of research activity at scene may not delay enrolled patient treatment, it is possible that delay required to complete additional research activity steps after patient handover may prolong the turnaround phase.

In summary, we completed a large prehospital stroke trial involving a simple-to-administer intervention across multiple ambulance services. The time from onset to randomisation and modest distances travelled support the applicability of future large-scale paramedic-delivered ambulance-based stroke trials in urban and rural locations.

Nevertheless, prehospital time intervals and distances from scene-to-hospital varied by ambulance service and this was, at least in part, explained by the type of urban versus rural population. Although our results may not be generalisable to all ambulance service settings, they do inform future developments in ambulance-based stroke care and provide support to the deliverability of future large-scale multicentre pre-hospital paramedic-delivered ambulance-based acute stroke trials.

**Author affiliations**
[1]Division of Mental Health and Clinical Neuroscience, University of Nottingham Faculty of Medicine and Health Sciences, Nottingham, UK
[2]Leicester, Leicestershire & Rutland Division, East Midlands Ambulance Service NHS Trust, Nottingham, UK
[3]Stroke, University Hospitals Birmingham NHS Foundation Trust, Birmingham, UK
[4]Division of Paramedic Science, School of Health and Social Work, University of Hertfordshire, Hatfield, UK
[5]School of Health and Social Care, University of Lincoln, Lincoln, UK

**Acknowledgements** We thank the patients who participated in this trial and their relatives, the clinical and research teams of the various ambulance services and hospitals and the paramedics who participated, recruited and treated the patients. The following ambulance service colleagues are acknowledged who contributed to trial implementation, data collection and revision of the draft paper: Roderick Johnson, paramedic clinical operations manager (East Midlands Ambulance Services (EMAS)); Debbie Shaw, clinical audit and research manager (EMAS); Robert Spaight, head of research and audit (EMAS); Larissa Prothero, research paramedic (East of England Ambulance Services (EEAS)); Theresa Foster, research manager (EEAS); Rachael T Fothergill, research manager (London Ambulance Services (LAS)); Heather Cole, research paramedic (LAS); Joanna Lazarus, research paramedic (LAS); Helen Werts, research paramedic (LAS); Neil Thomson, medical director (LAS); Helen Pocock, senior research paramedic (South-Central Ambulance Services (SCAS)); Kurtis Poole, research paramedic (SCAS); Ed England, research manager (SCAS); Maria Robinson, research manager (South-West Ambulance Services (SWAS)); Katherine Zorab, quality improvement paramedic (SWAS);

Adrian South, Clinical Director (SWAS); Catrin Convery, research paramedic (Welsh Ambulance Services (WAS)); Tim Pearce, paramedic team leader (WAS); Nigel Rees, head of research and development (WAS); Joshua Miller, research paramedic (West-Midlands Ambulance Services (WMAS)); Imogen Gunson, research paramedic (WMAS); Andrew Rosser, head of research (WMAS); Matthew Ward, consultant paramedic (WMAS); Kelly Hird, research paramedic (Yorkshire Ambulance Services (YAS)); Jacqui Crossley, assistant clinical director (YAS). We acknowledge support of the English National Institute for Health Research (NIHR) Clinical Research Network, Health and Care Research Wales, and that the coordination between multiple ambulance services and hospitals, and large recruitment would not have been possible without NIHR network support. A complete list of the Rapid Intervention with Glyceryl trinitrate in Hypertensive stroke Trial-2 (RIGHT-2) investigators is provided in the primary publication. We acknowledge Eivind Berge who contributed to this paper but sadly passed away prior to publication.

**Collaborators** The following are members of RIGHT-2 (Rapid Intervention with Glyceryl trinitrate in Hypertensive stroke Trial-2) trial steering committee and investigator group who critically revised the manuscript for important intellectual content: Mark Dixon, paramedic divisional senior manager (quality), Nottingham; Jason P Appleton, consultant neurologist, Birmingham; Polly Scutt, medical statistician, Nottingham; Lisa J Woodhouse, medical statistician, Nottingham; Lee J Haywood, database programmer, Nottingham; Harriet Howard, senior trial coordinator, Nottingham; Diane Havard, senior trial manager, Nottingham; Nikola Sprigg, consultant stroke physician, Nottingham; Tom Robinson, professor of stroke medicine, Leicester; Christopher Price, clinical reader in stroke medicine, Newcastle-upon-Tyne; Craig Anderson, professor of neurology and epidemiology, Beijing; Grant Mair, senior clinical lecturer in neuroradiology, Edinburgh; Else C Sandset, consultant neurologist, Oslo; Jeffrey Saver, professor of clinical neurology, Santa Monica; Christine Roffe, stroke physician, Stoke-on-Trent; Keith Muir, consultant neurologist, Glasgow; Kailash Krishnan, consultant neurologist, Nottingham; Joanna M Wardlaw, professor of applied neuroimaging, Edinburgh; Julia Williams, professor of paramedic science, Hatfield; A Niroshan Sirawardena, professor of primary and prehospital care, Lincoln; Philip M Bath, professor of stroke medicine, Nottingham.

**Contributors** PMB, also chief investigator and guarantor, and MD conceived the study. All authors contributed to the planning, design and conduct. LJH was responsible for data curation. PS and LJW supported with statistical analysis. All authors contributed to the reporting, analysis and interpretation of the results. MD and PMB led the writing of the manuscript with critical revision from JPA, PS, LJW, LJH, DH, JW and ANS.

**Funding** This work was supported by the British Heart Foundation (grant number CS/14/4/30972).

**Map disclaimer** The inclusion of any map (including the depiction of any boundaries therein), or of any geographical or locational reference, does not imply the expression of any opinion whatsoever on the part of BMJ concerning the legal status of any country, territory, jurisdiction or area or of its authorities. Any such expression remains solely that of the relevant source and is not endorsed by BMJ. Maps are provided without any warranty of any kind, either express or implied.

**Competing interests** JPA was funded in part by the British Heart Foundation (BHF) during the conduct of the study and is supported by the National Institute for Health Research (NIHR) West Midlands Health and Care Research Scholars Programme. CSA reports grants from National Health and Medical Research Council (NHMRC) of Australia, grants from Takeda, and personal fees from Takeda, Amgen and Boehringer Ingelheim outside of the submitted work. MD was funded by the BHF during the conduct of the study. TJE and AAM report grants from BHF during the conduct of the study. GM is supported by National Health Service Lothian Research and Development Office and reports grants from The Stroke Association (TSA), The Royal College of Radiologists. CIP reports grants from Nottingham University and BHF during the conduct of the study. TGR is an NIHR Senior Investigator, and reports grants from BHF during the conduct of the study. CR reports grants from NIHR Health Technology Assessment during the conduct of the study; personal fees from Allergan, Air Liquide, Merz, Boehringer, Bayer, Johnson & Johnson, Sanofi and Emtensor; non-financial support from European Stroke Conference and Trident; and other support from Firstkind Medical, Medtronic and Brainomix outside the submitted work. PMR reports grants from Wellcome Trust during the conduct of the study. ECS reports personal fees from Novartis and Bayer outside of the submitted work. JMW reports grants from the BHF during the conduct of the study; and grants from Medical Research Council, Chief Scientist Office, Leducq, EU H2020, TSA, BHF and Alzheimer's Society outside the submitted work. NSp reports grants from BHF and RCUK, during the conduct of the study. PMB is Stroke Association Professor of Stroke Medicine and is an NIHR Senior Investigator. He reports grants from BHF

during the conduct of the study; personal fees and other fees from Sanofi, Nestlé, DiaMedica, Moleac, Platelet Solutions, Phagenesis, and ReNeuron, outside the submitted work.

**Patient and public involvement** Patients and/or the public were involved in the design, or conduct, or reporting, or dissemination plans of this research. Refer to the Methods section for further details.

**Patient consent for publication** Not applicable.

**Ethics approval** RIGHT-2 was approved by the UK regulator (Medicines and Healthcare products Regulatory Agency, reference: 03057/0064/001–0001; Eudract 2015–000115–40) and national research ethics committee (IRAS: 167115) and was adopted by the National Institute for Health Research Clinical Research Network. Participants gave informed consent to participate in the study before taking part.

**Provenance and peer review** Not commissioned; externally peer reviewed.

**Data availability statement** Data are available upon reasonable request. Individual participant data are shared with the Blood pressure in Acute Stroke Collaboration and Virtual International Stroke Trials Archive.

**ORCID iDs**
Mark Dixon http://orcid.org/0000-0002-4036-3792
Lisa J Woodhouse http://orcid.org/0000-0002-4472-1999
Julia Williams http://orcid.org/0000-0003-0796-5465
Aloysius Niroshan Siriwardena http://orcid.org/0000-0003-2484-8201

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
