## [Reviewer comments · BMJ Open]

ARTICLE DETAILS

TITLE (PROVISIONAL)	Time intervals and distances travelled for pre-hospital ambulance stroke care: data from the randomised-controlled ambulance-based Rapid Intervention with Glyceryl trinitrate in Hypertensive stroke Trial-2 (RIGHT-2)
AUTHORS	Dixon, Mark; Appleton, Jason; Scutt, Polly; Woodhouse, Lisa; Haywood, Lee; Havard, Diane; Williams, Julia; Siriwardena, Aloysius; Bath, Philip; RIGHT-2 Investigators

VERSION 1 – REVIEW

REVIEWER	Zhao, Henry The Royal Melbourne Hospital, Department of Neurology
REVIEW RETURNED	27-Feb-2022

GENERAL COMMENTS	After reviewing responses to the previous review I believe the authors have addressed the major issues, although I have some further minor comments: 1. The authors should avoid country specific terms such as “999-call” and replace it with generic terminology that will be understood internationally2. I am still not sure that the direct consent model mentioned by the authors is “novel” (unless it is novel for this sort of trial in the UK). Globally in pre-hospital stroke research the push is in the opposite direction, that is to allow for waiver or deferral of consent as stroke is a time-critical emergency. In addition, there are real questions of whether consent is truly “informed” with patients and families under enormous stress and time constraints to make a decision. I also do not agree with the assertion by the authors that the direct consent model has less selection bias than waiver/deferred consent as there are some patient groups less likely to be consented in this way, such as those in culturally and linguistically diverse communities.3. The authors might like to consider in Table 2 breaking the 61-120 timepoint data into 61-90 and 91-120 as the 90-min timepoint is of some interest in the stroke community4. Regarding the Google maps images in supplementals – what do the pins mean? It may be also worthwhile checking there are no copyright issues before publication.
---

REVIEWER	Bury, Gerard University College Dublin, General Practice
REVIEW RETURNED	01-Mar-2022

GENERAL COMMENTS	This group has published key reports from a landmark pre-hospital interventional study on stroke. This study explores detailed background data from that study - however, much of this data
---

	appears to be audit type data and overall it's unclear what it adds to the earlier publications. Comments:  1. The aims of the study are unclear but appear to focus on detailed descriptive time period data from participating services. The data is not analysed in ways which compare it with baseline data for these services or other key items and it's unclear what significance can be attached to straightforward time periods. The finding that large numbers of potential stroke patients can be assessed and (potentially) recruited within a relevant time period for stroke is important but has been reported in earlier studies. 2. The consent process for 10% of the patients involved the paramedic giving consent on behalf of the patient. Although the REC appears to have approved this, it would be interesting and important to explain how this sits with the legal position on consent. 3. There is no information on handover delays or what proportion of patients were pre-alerted to the receiving centre. Did pre-alerting occur in any specific groups and did it affect the time windows? 4. Were there differences between the staff members who were trained and those who were not or between those who recruited patients and those who did not? The potential for issues such as experience, seniority, age or gender etc to impact on participation or recruitment would be interesting data and in turn might have implications for use of time. 5. The variations between services in recruitment numbers are striking and not immediately explained. Perhaps with more recruitment and exposure to cases, time periods on scene might become quicker? For those services with few cases, were delays equivalent to those with many? 6. The numbers of stroke mimics is high and differs widely between services. However issues such as hypoglycemia or post-ictal patients were exclusion criteria. Is more detailed information available on the nature of the stroke mimics included in the trial? Did any of these groups differ from the population in terms of time windows and why? 7. Overall, the study raises a number of questions which may fall within the ability of the authors to address and which may add usefully to the value of this paper. However, as it currently stands, the descriptive data included has limited impact.
--	---

VERSION 1 – AUTHOR RESPONSE

Reviewer: 1

Dr. Henry Zhao, The Royal Melbourne Hospital

Many thanks for taking the time to peer review this paper.

Comments to the Author:

After reviewing responses to the previous review I believe the authors have addressed the major issues, although I have some further minor comments:

1. The authors should avoid country specific terms such as “999-call” and replace it with generic terminology that will be understood internationally

Thank you – this terminology throughout the paper has been amended to emergency call to support with wider international understanding.

2. I am still not sure that the direct consent model mentioned by the authors is “novel” (unless it is novel for this sort of trial in the UK). Globally in pre-hospital stroke research the push is in the opposite direction, that is to allow for waiver or deferral of consent as stroke is a time-critical emergency. In addition, there are real questions of whether consent is truly “informed” with patients and families under enormous stress and time constraints to make a decision. I also do not agree with the assertion by the authors that the direct consent model has less selection bias than waiver/deferred consent as there are some patient groups less likely to be consented in this way, such as those in culturally and linguistically diverse communities.

Thank you, for your thoughts. With your comments in mind and those of Reviewer 2, this section has been re-worked to aid clarity in understanding the consent process and methods utilised within this study. The term ‘selection bias’ has been removed and reworked to ensure our position is articulated in a clearer way.

3. The authors might like to consider in Table 2 breaking the 61-120 timepoint data into 61-90 and 91-120 as the 90-min timepoint is of some interest in the stroke community

Thank you for this valid comment, the time periods have been amended as suggested and included within Table 2.

4. Regarding the Google maps images in supplementals – what do the pins mean? It may be also worthwhile checking there are no copyright issues before publication. Pins on the maps have been identified. Maps have also been amended to align with the required attribution guidelines as noted on the website:

<https://about.google/brand-resource-center/products-and-services/geo-guidelines/#required-attribution>

Reviewer: 2

Dr. Gerard Bury, University College Dublin

Thank you for your time and questions posed from review of this manuscript. We have reviewed the manuscript and responded to your comments below.

Comments to the Author:

This group has published key reports from a landmark pre-hospital interventional study on stroke. This study explores detailed background data from that study - however, much of this data appears to be audit type data and overall it's unclear what it adds to the earlier publications. Comments:

1. The aims of the study are unclear but appear to focus on detailed descriptive time period data from participating services. The data is not analysed in ways which compare it with baseline data for these services or other key items and it's unclear what significance can be attached to straightforward time periods. The finding that large numbers of potential stroke patients can be assessed and (potentially) recruited within a relevant time period for stroke is important but has been reported in earlier studies.

Thank you. The aim of this paper is to add to the body of literature that supports the development of ambulance-based research with the focus being across multiple organisations in light of the experience from RIGHT-2. The existing ambulance-based trials in stroke have all been single centre, but we hope that this paper highlights the success and confirms the feasibility that multi-centre ambulance-based research is indeed possible albeit highlighting some challenges for consideration for future researchers.

2. The consent process for 10% of the patients involved the paramedic giving consent on behalf of the patient. Although the REC appears to have approved this, it would be interesting and important to explain how this sits with the legal position on consent.

Thank you for your comments and in consideration with the comments from Reviewer 1 – this section has been reviewed and revised which we hope aids clarity regarding the processes applied.

3. There is no information on handover delays or what proportion of patients were pre-alerted to the receiving centre. Did pre-alerting occur in any specific groups and did it affect the time windows?

Thank you - turnaround times and handover times are acknowledged as a limitation – these were not collected; however, it is acknowledged that particularly in the current health service climate, handover and turnaround times are under increasing scrutiny so as a recommendation we discuss this as a necessity for future prehospital research where timings are collated.

Pre-alerting occurred in line with ambulance service guidance at the time of the trial and supports the rationale for including patients within a 4-hour window of symptom onset.

5. Were there differences between the staff members who were trained and those who were not or between those who recruited patients and those who did not? The potential for issues such as experience, seniority, age or gender etc to impact on participation or recruitment would be interesting data and in turn might have implications for use of time.

Thank you for this comment and we have considered this at length. The characteristics of Paramedics is planned to be explored in future publication, however the focus and scope of this paper remains with the timings and distances in view of multi-centre research.

6. The variations between services in recruitment numbers are striking and not immediately explained. Perhaps with more recruitment and exposure to cases, time periods on scene might become quicker? For those services with few cases, were delays equivalent to those with many?

Thank you – we acknowledge that in part due to delays in individual centre set-up and a low recruitment rate seen in the first few months of the trial, further ambulance services and hospitals were set up throughout the recruitment phase with the final ambulance service joining for only a four-month period. The recruitment rate of patients per paramedic should help in understanding this further. As an additional complexity, once some stroke centres had received their contracted target number of participants they switched off to receiving patients from ambulance services. The reliance of dual centres to recruit is challenging and acknowledged here and has been added to the discussion.

Indeed, the additional stroke education around the trial, familiarity with recruitment processes is important to review and will be reviewed on a paramedic-by-paramedic basis for a future publication.

6. The numbers of stroke mimics is high and differs widely between services. However issues such as hypoglycemia or post-ictal patients were exclusion criteria. Is more detailed information available on the nature of the stroke mimics included in the trial? Did any of these groups differ from the population in terms of time windows and why?

The recent publication by RIGHT-2 Investigators on mimics has been referred to within the text – thank you.

7. Overall, the study raises a number of questions which may fall within the ability of the authors to address and which may add usefully to the value of this paper. However, as it currently stands, the descriptive data included has limited impact.

Thank you for your valued comments. We hope the revisions made to this paper assist with improving the impact and value to the readership.

VERSION 2 – REVIEW

REVIEWER	Bury, Gerard University College Dublin, General Practice
REVIEW RETURNED	19-Aug-2022
GENERAL COMMENTS	Many thanks for revised submission, which addresses issues raised.